# Iriflophenone-3-C-β-d Glucopyranoside from *Dryopteris ramosa* (Hope) C. Chr. with Promising Future as Natural Antibiotic for Gastrointestinal Tract Infections

**DOI:** 10.3390/antibiotics10091128

**Published:** 2021-09-18

**Authors:** Muhammad Ishaque, Yamin Bibi, Samha Al Ayoubi, Saadia Masood, Sobia Nisa, Abdul Qayyum

**Affiliations:** 1Department of Botany, PMAS-Arid Agriculture University Rawalpindi, Rawalpindi 46300, Pakistan; mishaque270@gmail.com (M.I.); dryaminbibi@uaar.edu.pk (Y.B.); 2College of Humanities and Sciences, Prince Sultan University, Rafha Street, Riyadh 11586, Saudi Arabia; sayoubi@psu.edu.sa; 3Department of Statistics & Mathematics, PMAS-Arid Agriculture University Rawalpindi, Rawalpindi 46300, Pakistan; 4Department of Microbiology, The University of Haripur, Haripur 22620, Pakistan; sobia@uoh.edu.pk; 5Department of Agronomy, The University of Haripur, Haripur 22620, Pakistan

**Keywords:** *Dryopteris ramosa*, ethnomedicinal uses, iriflophenone-3-C-β-D glucopyranoside, GIT infection, antibacterial, cytotoxic potential

## Abstract

Ethnopharmacological approaches provide clues for the search of bioactive compounds. *Dryopteris ramosa* (Hope) C. Chr. (plant family: Dryopteridaceae) is an ethnomedicinal plant of the Galliyat region of Pakistan. The aqueous fraction (AqF) of *D. ramosa* is being used by inhabitants of the Galliyat region of Pakistan to treat their gastrointestinal tract ailments, especially those caused by bacteria. The aims of the present study were as follows: (i) to justify the ethnomedicinal uses of the AqF of *D. ramosa*; (ii) to isolate a bioactive compound from the AqF of *D. ramosa*; and (iii) to evaluate the antibacterial and cytotoxic potential of the isolated compound. Column chromatography (CC) techniques were used for the isolation studies. Spectroscopic techniques (UV–Vis, MS, 1&2D NMR) were used for structural elucidation. The agar-well diffusion method was used to evaluate the antibacterial potential of “i3CβDGP” against five bacterial strains, and compare it with the known antibiotic “Cefixime”. The brine shrimp lethality test (BSLT) was used for cytotoxic studies. The AqF of *D. ramosa* afforded “iriflophenone-3-C-β-D glucopyranoside (i3CβDGP)” when subjected to LH20 Sephadex, followed by MPLC silica gel_60_, and purified by preparative TLC. The “i3CβDGP” showed a strong potential (MIC = 31.1 ± 7.2, 62.5 ± 7.2, and 62.5 ± 7.2 µg/mL) against *Klebsiella pneumoniae*, *Staphylococcus aureus*, and *Escherichia coli*, respectively. On the other hand, the least antibacterial potential was shown by “i3CβDGP” (MIC = 125 ± 7.2 µg/mL), against *Bacillus subtilis*, in comparison to Cefixime (MIC = 62.5 ± 7.2 µg/mL). The cytotoxicity of “i3CβDGP” was significantly low (LD_50_ = 10.037 ± 2.8 µg/mL) against *Artemia salina* nauplii. This study not only justified the ethnomedicinal use of *D. ramosa*, but also highlighted the importance of ethnomedicinal knowledge. Further studies on AqF and other fractions of *D. ramosa* are in progress.

## 1. Introduction

Chemical compounds obtained from plants present a great structural diversity and enact important precursor compounds for the development of new restorative agents [1]. The evolutionary selection pressures compel plants to produce diverse phytochemicals. This structural diversity of phytochemicals enables them to interfere with different biological components of the cells, for specific purposes. These bioactivities form the evidence by which some of them become efficient drugs in a wide range of remedying indications [2].

In the recent era, for the search of bioactive natural constituents from plants, ethnobotanical and ethnopharmacological knowledge is being used as a starting point. A large number of pharmaceutical natural products have been obtained from the plants, on the basis of this empirical and traditional knowledge [3].

*Dryopteris ramosa* (Hope) C. Chr. is a common fern plant (Figure 1) of the Galliyat region of Pakistan. It belongs to the plant family Dryopteridaceae. Many ethnobotanists and ethnopharmacologists have reported the ethnomedicinal use of *Dryopteris ramosa*. For instance, the aqueous fraction (leaves) of *Dryopteris ramosa* has been used as gastrointestinal tonic [4], to cure stomach pain [5], for the treatment of gastric ulcers and constipation [6,7]. *D. ramosa* has been reported to have antimicrobial and anticancer potential [8]. In the recent past, [9] has discussed the antioxidant potential of *D. ramosa*. The crude extract and various fractions of *D. ramosa* showed strong antioxidant potentials. The total phenolics and flavonoid contents were determined in a previous study [9].

In this study, we focused on the aqueous fraction of *D. ramosa* (leaves/fronds), because in traditional medicines, the polar fraction of *D. ramosa* is being used to cure gastrointestinal tract (GIT) disorders. Further, it is to validate the traditional use of the aqueous fraction of *Dryopteris ramosa* for GIT disorders. In this research, the isolation of a pure bioactive compound and its structure elucidation has been carried out by using chromatographic and spectroscopic techniques. Furthermore, in vitro bioactivities, including the antimicrobial and cytotoxic potential of a pure isolated compound from the AqF of *D. ramosa*, have been discussed. The bacteria that are commonly responsible for GIT disorders were used in this study. Table 1 shows the list of the bacteria selected for the present study, and their importance in gastrointestinal tract (GIT) disorders. A cytotoxic evaluation has been carried out to determine whether the isolated pure compound can be used as a future antibiotic or not. This present research article is a novel work, as it is the first report on the antimicrobial and cytotoxic potential of an isolated compound from the aqueous fraction of *D. ramosa*. Furthermore, this is the first report of the isolation of antibacterial phytochemicals from *D. ramosa*, to the best of our knowledge.

## 2. Materials and Methods

### 2.1. Plant Material Used

Fronds of *Dryopteris ramosa* (locally known as Pakha or pakhi) were collected (2 kg fresh weight) from Galliyat region (near Patriata/New Murree) of Pakistan according to the “Guidelines in the Conservation on Medicinal Plants (1993)”. The plant was identified by an expert taxonomist (Dr. Rehmatullah Qureshi) at PMAS-Arid Agriculture University Rawalpindi, Pakistan, and voucher specimen (MI-934/15) was deposited in herbarium of Quaid-I-Azam University, Islamabad for future reference.

### 2.2. Formation of Crude Methanolic Plant Extract

Collected leaves of selected plant were dried and powder was formed in a crushing mill. By using cold maceration technique, crude methanolic extract (CME) was formed. Extract was rotary evaporated and stored at 4 °C until further use.

### 2.3. Formation of Aqueous Fraction of D. ramosa

The crude methanolic extract (10 g) was subjected to the polarity-based solvent–solvent fractionation. Fractionation procedure of CME in the form of a schematic diagram is presented in Figure 2. The CME (10) was dissolved in equal volume (50 mL each) of water and n-Hexane in a fractionation flask and allowed to stand. After 10 min two layers were evidenced in the fractionation flask. Upper layer was separated, rotary evaporated and stored as “nHF”. The lower layer (polar) was mixed with 50 mL of chloroform, shook and allowed to stand in fractionation flask for 20 min. Again two layers were formed, but as the density of chloroform is higher than water, this time lower layer was organic layer (non-polar). The organic layer was separated, rotary evaporated and stored as “CF”. Once again, the upper polar layer was mixed with ethyl acetate (50 mL), shook vigorously and allowed to stand for an hour. Two layers were formed along with some sediment material. The layers were separated, rotary dried and labeled as “EAF” (upper layer) and “AqF” (lower layer). The sediment material was dissolved and soluble in methanol. It was labeled as “MF” after rotary evaporation of methanol. Every step was repeated three times. All the fractions obtained were stored at 4 °C until further use.

### 2.4. Isolation of Bioactive Compounds from AqF of D. ramosa

Two grams of aqueous fraction (AqF) of *D. ramosa* was chromatographed over Sephadex LH20 eluted isocratic with methanol. A total of 20 elutions (50 mL of each) were collected, and based on TLC analysis they were pooled into four groups. On the account of MIC against bacterial strains, group 2 (139 mg) was subjected to CC over Sephadex LH20 (120 × 2 cm) eluted (18 fractions, 20 mL of each) with methanol. Fractions were pooled into two groups and group 2a (83.71 mg) was purified by MPLC over silica gel 60, 40–60 µm particle size, incurring 39.15 mg of pure compound. TLC analyses were carried out on silica gel 60 F_254_ plates developed in the mixture of n-butanol:acetic acid:water (4:1:5; *v*/*v*/*v*).

### 2.5. Structure Elucidation of Isolated Compound

#### 2.5.1. UV–Vis Spectroscopy

HPLC analyses were conducted on Agilent 1100 series with UV-DAD detector, Hypersil BDS-C18 column (250 × 4.6 mm, particle size = 5 µm). The solvent system was methanol in aqueous buffer (15 mM ortho-H_3_PO_4_ and 1.5 mM Bu_4_NOH), flow rate 1 mL/min., injection volume 10 µL and linear gradient starting from 20% MeOH to 90% at 17 min to 100% at 20 min kept for 8 min, wavelength scanned between 200 nm and 600 nm.

#### 2.5.2. Mass Spectroscopy (MS Analysis): Instrument: HR-TOF Mass Spectrometer (maXis, Bruker Daltonics)

Method: Direct infusion electrospray ionization (ESI) with positive mode. Spectra range: 100–2500 *m*/*z*, capillary voltage: 4 kV, capillary current: 30–50 nA, nitrogen temperature: 180 °C, flow rate: 4.0 L min^−1^, and N_2_ nebulizer gas pressure: 0.3 bar.

#### 2.5.3. Nuclear Magnetic Resonance (NMR) Spectroscopy

Instrument: Bruker Avance II 400 (resonance frequencies 400.13 MHz for ^1^H NMR and 100.63 MHz for ^13^C NMR) equipped with a 5 mm observe broadband probe head (BBFO) with z-gradients at room temperature with standard Bruker pulse programs. Solvent: 0.6 mL of MeO-d_4_ (99.8% D). ^1^H NMR: 32 k complex data points and apodized with a Gaussian window function (lb = −0.3 Hz and gb = 0.3 Hz), ^13^C-jmod spectra with WALTZ16, ^1^H decoupling was acquired using 64 k data points. Signal-to-noise enhancement was achieved by multiplication of the FID with an exponential window function (lb = 1 Hz). All two-dimensional experiments (COSY, HSQC, HMBC) were performed with 1 k × 256 data points. Measurement temperature: 298 K ± 0.05 K. Residual CD_3_OD was used as internal standard for 1 H (δH 3.34) and CD_3_OD for 13 C (δC 49.0) measurements.

### 2.6. Evaluation of Antibacterial Properties

#### 2.6.1. Microorganisms Used

The isolated compound from the aqueous extract of *D. ramosa* was tested against five bacterial strains including two Gram-positive and three Gram-negative strains. Two Gram-positive bacteria species were as follows: *Bacillus subtilis* (ATCC 6633) and *Staphylococcus aureus* (ATCC 6538) while three Gram-negative bacteria were as follows: *Klebsiella pneumoniae* (ATCC 700603), *Salmonella enterica* subsp. *Enterica serovar* Setubal (ATCC 19196) and *Escherichia coli* (ATCC 25922).

#### 2.6.2. Control Experiments

Cefixime (antibiotic) was used as standard and positive control while Dimethyl sulfoxide (DMSO) was used as negative control.

#### 2.6.3. Preparation of Inoculum

Authentic strains of bacteria were cultivated in nutrient broth agar at 37 ± 0.3 °C for 24–48 h. This culture was maintained throughout the study in agar slanted suspension at 4 °C. Bacterial culture were suspended in 0.85% NaCl solution and adjusted to a turbidity 0.5 MacFarlan standard (10^8^ ± 0.5 CFU).

#### 2.6.4. Antibacterial Assay

The crude methanolic extract of *D. ramosa* and the isolated compound were then evaluated for antimicrobial activity using standard antibacterial assay “agar-well diffusion method” as described by [18] with some modifications. Sterilized nutrient agar plates were rinsed with 1 mL of bacterial suspension. After 10 min, a sterile cork borer (8 mm diameter) was then used to cut equidistant wells on the surface of the agar. Test compound (20 µL) of various concentrations (10, 100, 500 and 1000 µg/mL) was poured in each well. The plates were incubated for 24 h under aerobic conditions, at 37 °C. After 24 h of incubation, coalescent bacterial growth was observed. Inhibition of the bacterial growth around each well was recorded in millimeters (mm). Reference positive control (Cefixime—a well-known antibiotic) was used. Tests were conducted in triplicates.

#### 2.6.5. Determination of Minimum Inhibitory Concentration (MIC)

MIC values were defined as the lowest concentration of each natural product, which completely inhibited microbial growth. The results were expressed in milligrams per milliliter [19]. MIC was determined as describe by [20] with some modifications.

The MIC value of compound and CME were determined by 10 serial dilutions (500, 250, 125, 62.5, 31.25, 15.62, 7.81, 3.95, 1.98, 0.99 and 0.49 µg/mL). Equal volumes of test compound (dilution) and nutrient broth were mixed in each test tube. Inoculum (20 µL, 108 cfu) was also added in each tube. The tubes were incubated aerobically at 37 °C for 24 h. Two control tubes for each dilution were maintained. One included test compound (each dilution) and growth medium while second tube contained growth medium, physiological saline and inoculum. MIC was determined as the lowest concentration of the compound permitting no visible growth (no turbidity) when compared with the control tubes [20].

### 2.7. Determination of Cytotoxic Potential of Isolated Pure Compound from AqF of D. ramosa

With some modifications, brine shrimp lethality test (BSLT) was carried out to evaluate the cytotoxic potential of i3βDGP as described by [21].

#### 2.7.1. Hatching of Brine Shrimps

Hatching of brine shrimp (*Artemia salina*) eggs were carried out in artificial sea water prepared from commercial sea salt 38 g/L and with aeration for 3 h using air pump. *Artemia salina* hatched after 24 h as shrimps nauplii and was ready for the assay. A source of light (lamp/light bulb) was used to attract hatched nauplii closer to the wall of container.

#### 2.7.2. Bioassay—BSLT Procedure

Pure isolated compound (10 mg/mL) was dissolved in 100% DMSO as stock solution. Less than 1.25% DMSO was used to prepare each dilution (*v*/*v*) to achieve the maximum tolerable concentration of working solution as described by [22]. From the stock solution, 5, 10, 100, 300 and 600 µg/mL concentrations were prepared with artificial and well-aerated sea water in glass vials with 5 mL final volume. The largest concentration 300 µg/mL had 1% DMSO. Potassium dichromate (K_2_Cr_2_O_7_) was used as positive control while well-aerated artificial sea water was used as negative control. The phototropic nauplii (24–36 h old) were collected with the help of pasture pipette and 20 nauplii were introduced into each concentration/vial and incubated for 24 h at room temperature. After 24 h of incubation, the dead and alive nauplii were counted in each tube and the percentage death of nauplii was determined by using following equation:Percentage of Death (%) = (Total nauplii − Alive nauplii) × 100%/Total nauplii(1)

The cytotoxicity was calculated by regression line equation in the form of LD_50_ by using MS Excel 2010 software.

### 2.8. Statistical Analysis

All experiments were performed in triplicates and results were presented in ± standard deviation (SD). Analysis of variance (ANOVA) was used to determine the correlation and main effects of various concentrations of plant extract, fractions and isolated pure compound at *p* = 0.05 by using SPSS 16.0. LD_50_ was calculated by regression line equations in MS excel 2010. Chemical structures were drawn by using Chem Draw pro 8 free software.

## 3. Results

### 3.1. Extraction and Fractionation

After maceration, about 10 g of CME (2.22% of total dry weight) was obtained. This CME afforded nHF = 3.6 g, CF = 1.1 g, EAF = 1.8 g, AqF = 2.9 g, and MF = 0.6 g (Figure 3).

### 3.2. Spectroscopic Analysis of Isolated Pure Compound from AqF of D. ramosa

From the AqF of *D. ramosa*, 39.15 mg of pure compound was isolated, as stated in Section 2.3. The isolated compound was identified as iriflophenone-3-C-β-D glucopyranoside. In lieu, for the identification and structure elucidation of the isolated pure compound from the AqF of *D. ramosa*, the spectroscopic data are as follows:

UV–Vis spectroscopy: detector: DAD; RT: 6.467 min; λ _max_ = 298nm (band-I) and 210 nm (band-II). ^1^HNMR (400 MHz, MeOD-*d4*, δ, ppm): 7.61 (H-13, s, 1H), 7.59 (H-9,s, 1H), 6.78 (H-12,s, 1H), 6.75 (H-10, s, 1H), 5.94 (H-3, s, 1H), 4.88 (H-1’, d, *J* = 9.88, 1H), 3.89 (H-2’, m, 1H), 3.84 (H-6’, d, *J* = 1.96, 2H), 3.48 (H-4’, m, 1H), 3.44 (H-3’, m, 1H), and 3.35 (H-5’, m, 1H). ^13^CNMR (MeOD- *d*,): δ 198.89 (C-7), 163.42 (C-6), 162.90 (C-11), 161.44 (C-4), 161.07 (C-2), 133.26 (C-8), 132.82 (C-9), 132.82 (C-13), 115.50 (C-10), 115.50 (C-12), 107.17 (C-1), 104.67 (C-5), 94.61 (C-3), 82.55 (C-5’), 79.93 (C-3’), 76.61 (C-1’), 73.74 (C-2’), 71.46 (C-4’), and 62.48 (C-6’). EIMS ESI-TOF (+) = 409 (M + H) *m*/*z*, 246 *m*/*z* (M-163 + H), and other fragments include the following: 392 *m*/*z* (M-17 + H), 288 *m*/*z* (M-OH-105 + H), 126 *m*/*z* (M-163-121 + H). Molecular weight: 408 amu, and molecular formula: C_19_H_20_O_10_.

### 3.3. Antibacterial Potential of “Iriflophenone-3-C-β-D Glucopyranoside”

The antibacterial potential of the isolated compound (i3βDGP) from the AqF of *D. ramosa* and Cefixime (antibiotic/control), against different strains of bacteria, is presented in Table 2. A comparison of the MIC values of i3βDGP and Cefixime is illustrated in Figure 4, against various bacterial pathogens of human GIT.

### 3.4. Cytotoxic Properties of Iriflophenone-3-C-β-D Glucopyranoside

The cytotoxicity of the isolated compound (i3βDGP) and Cefixime was evaluated through BSLT. Potassium dichromate (K_2_Cr_2_O_7_) was used as a control. The mean percentage lethality against each concentration is presented in Figure 5. The LD_50_ (in µg/mL) was calculated by the regression equation. A low LD_50_ (µg/mL) value means that the compound is cytotoxic. LD_50_ of i3βDGP is 10.037 ± 2.89 µg/mL (Figure 5), which is high, so i3βDGP may not be considered as a cytotoxic compound.

## 4. Discussion

The traditional knowledge and discovery of a bioactive drug always goes hand in hand. The traditional ethno-knowledge not only helps in the selection of plant material for investigation, but also provides a clue in the search for effective and safer natural compounds. Medicinal plants play a vital role in the discovery and development of new drugs [23]. In Pakistan, the indigenous knowledge and medicinal plant are widespread in different tribes and localities [24]. *D. ramosa* (Figure 1) is an important ethnomedicinal plant of the Galliyat region of Pakistan. The aqueous fraction (AqF) of *D. ramosa* is being used as gastrointestinal tonic [4], and to cure GIT disorders, as mentioned in the Section 1 of this article [5,6,7]. Most of the GIT disorders are because of bacterial infection. A list of bacteria that are related to GIT disorders is given in Table 1.

In the quest of effective antibacterial compounds, the AqF of *D. ramosa* is subjected to column chromatography, which afforded 39.15 mg of pure compound. This compound was identified by spectroscopic techniques. The ^1^H and ^13^C NMR data of this compound are in accordance with the previously published data of this compound from *Cyclopia subternata* vogel (honey bush) [25]. The 2D NMR (gs HSQC and gs HMBC) plots are given in Figure 6a,b. Correlation spectroscopy/COSY (2D NMR) determined the correlation between coupled protons. A point of entry into the COSY spectrum is one of the keys to predicting information from it accurately. The relation between coupling protons is determined by diagonal lines (correlation peaks and COSY spectrum). The H-COSY spectrum (Figure 6a) of the isolated compound “i3CβDGP” suggested that H-1/(δ 4.88 ppm, d, *j* = 9.88) is coupled with the proton of H-2/(δ 3.89, m). Similarly, H-13 (δ 7.61, s) was coupled with H-12 (δ 6.78, s), and H-10 (δ 6.75, s) with H-9 (δ 7.59, s). The heteronuclear multiple-bond correlation (HMBC) experiment was developed to assist in the identification (correlation) of proton nuclei with carbon nuclei that are separated by more than one bond. The HMBC plot of “i3CβDGP” was given in Figure 6b. Three bond couplings were observed between H-3 (δ 5.94 ppm), C-1 (δ 107.17 ppm), and C-5 (δ 104.67 ppm). Similarly, a 3-*j*-CH correlation was noticed between H-9 (δ 7.59 ppm), C-7 (δ 198.89 ppm), and C-11 (δ 162.90 ppm), as well as H-10 (δ 6.75 ppm), C-12 (δ 115.50 ppm), and C-8 (δ 133.26 ppm). Abundant 3-*j*-CH long-range correlation signals were observed with cross peak H-1^/^(δ 4.88 ppm) to C-3^/^(δ 79.93 ppm), C-5^/^(δ 82.55 ppm), C-6 (δ 163.42 ppm), and C-4 (δ 161.44 ppm). The TOF-ESI/MS (Figure 7b) showed a peak of 288 amu (M + H-120), which is typical of a C-glycoside linkage. By using LC-MS/MS, [26] has reported similar observations. Based on these observations and comparisons of the present data from the literature, the compound was identified as “iriflophenone-3-C-β-D- glucopyranoside (i3CβDGP)” (Figure 7a). This is the first report of the isolation of “iriflophenone-3-C-β-D- glucopyranoside” from the fern genus *Dryopteris* (and so to *D. ramosa*), to the best of our knowledge.

The antibacterial potential of the isolated pure compound (“i3CβDGP”) against a selected strain of bacteria that is responsible for GIT disorders in humans is presented in Table 2. Cefixime, a well-known antibiotic, was used as a control, while distilled water was used as a blank. The MIC value was taken as the lowest concentration of “i3CβDGP” and cefixime that inhibited any visible bacterial growth. As presented in Table 2, “i3CβDGP” was the most effective against *Klebsiella pneumoniae* (MIC = 31.1 ± 7.2 µg/mL), while the same compound has shown minimum antibacterial potential against *Bacillus subtilis* (MIC = 125 ± 7.2 µg/mL). However, it was notable from Table 2 that the MIC value of “i3CβDGP” ranged between 31.1 and 125 µg/mL. In comparison with the known antibiotic “Cefixime”, “i3CβDGP” showed a similar MIC value for *Staphylococcus aureus*, *Klebsiella pneumoniae*, and *Escherichia coli*, but for *Salmonella enterica* and *Bacillus subtilis*, the MIC of “i3CβDGP” was greater than Cefixime (Figure 4). This indicated the promising future for “i3CβDGP” as an antibacterial natural compound. The antimicrobial properties of flavonoid glycosides have been reported by many researchers [27,28,29]. This is the first report on the antibacterial potential of “i3CβDGP”, although “i3CβDGP” was reported for its anti-hyperglycemic activities by [30], and antioxidant properties [31]. In a previous study, [29] isolated a number of flavonoid glycosides with antibacterial properties. According to them [29], the mode of antibacterial activity of flavonoid glycosides is due to their ability to cause cell lysis and disruption of the plasma membrane upon membrane permeability.

The inhibition zone and the MIC (µg/mL) were compared and presented in Table 2. The inhibition zone caused by the CME of *D. ramosa* ranges between 66 ± 0.7 mm (*S. aureus*) and 10 ± 0.7 mm (*E. coli*). The MIC for the CME against *B. subtilis* was the highest (250 ± 7.2 µg/mL), while it was the lowest against *K. pneumoniae* (62.5 ± 7.2 µg/mL). The antibacterial activity of crude extract was less as compared to isolated pure compound (i3CβDGP). This might be due to the synergetic effects between compounds. A similar observation was reported by [32]. They found that crude ethyl acetate root extraction of *Terminalia laxiflora* was less potent to *P. aeruginosa* than punicalagin (isolated from root extract of *Terminalia laxiflora*). A recent study was carried out by [33]. They presented the antimicrobial potential of *Dryopteris ramosa*. According to them, “the water fraction produced a 12 mm inhibition zone against MRSA-10, as compared to 10 mm given by the standard and 6 mm zone against *S. aureus*. The crude extract was the most active, amongst all the fractions, against *P. aeruginosa*, where it exhibited a zone of 13 mm while the standard produced a zone of 11 mm. DMSO being the negative control did not inhibit the zones. The MIC was calculated for all the extracts. According to the results, the lowest MIC was shown by the crude extract (16 μg/mL) against *P. aeruginosa*, as compared to the standard Cefixime, with a MIC of 32 μg/mL. Similarly, against MRSA-10, the methanolic and water fraction have a MIC of 32 μg/mL, as compared to the MIC of Cefixime (64 μg/mL).

The isolated pure compound “i3CβDGP” was subjected to BSLT to evaluate its cytotoxicity, and to compare it with the cytotoxicity of “cefixime” and potassium dichromate (k_2_Cr_2_O_7_). The results (Figure 5) indicated that “i3CβDGP” was not a cytotoxic compound (LD_50_ = 10.037 ± 2.89 µg/mL). This is a positive property, in the sense that it may be used as a future antibiotic, without causing significant cytotoxic adverse effects on healthy human cells in human GIT or in the body. Toxicity evaluation of pure compounds or plant extracts is important, prior to their further development and commercialization. BSLT is an easy, bench-top and cost-effective preliminary test to evaluate cytotoxicity. This bioassay is being used by many researchers to determine the cytotoxicity of plant extracts and pure isolated compounds [34,35].

## 5. Conclusions

The aqueous fraction (AqF) of *D. ramosa* is being used to treat gastrointestinal tract (GIT) ailments by the local inhabitants of the Gallyat region of Pakistan. We have successfully isolated the potential antibacterial compound “iriflophenone-3-C-β D glucopyranoside (i3CβDGP)” from the AqF of *D. ramosa*. “i3CβDGP” did not show significant cytotoxicity in BSLT, but “i3CβDGP” showed strong antibacterial potential against the tested strain of bacteria. This study not only justified the use of the AqF of *D. ramosa*, by the inhabitants of the Gallyat region of Pakistan, but also highlighted the importance of ethnomedicinal knowledge. Further studies on the AqF and other fractions of *D. ramosa* are in progress in our laboratory.

## Figures and Tables

**Figure 1 antibiotics-10-01128-f001:**
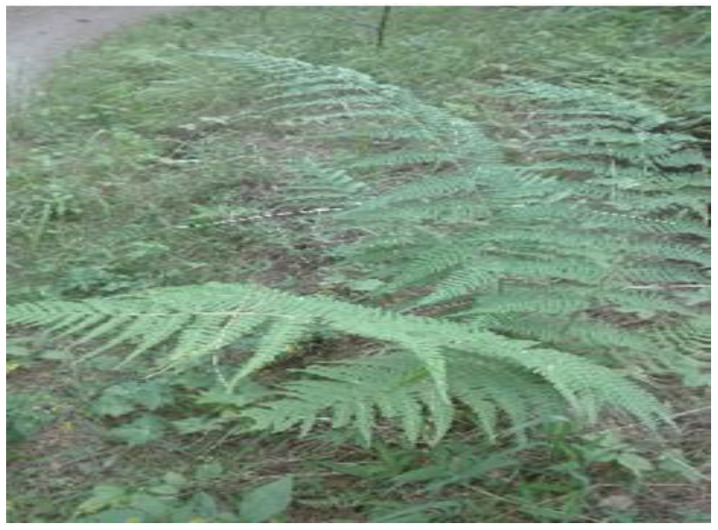
*Dryopteris ramosa* (Hope) C. Chr., growing at latitude 33°52′41.58″ N and longitude 73°7′8.47″ E (longitude and latitude were determined by using Google Earth pro 2002 free software). The photograph was taken by Muhammad Ishaque.

**Figure 2 antibiotics-10-01128-f002:**
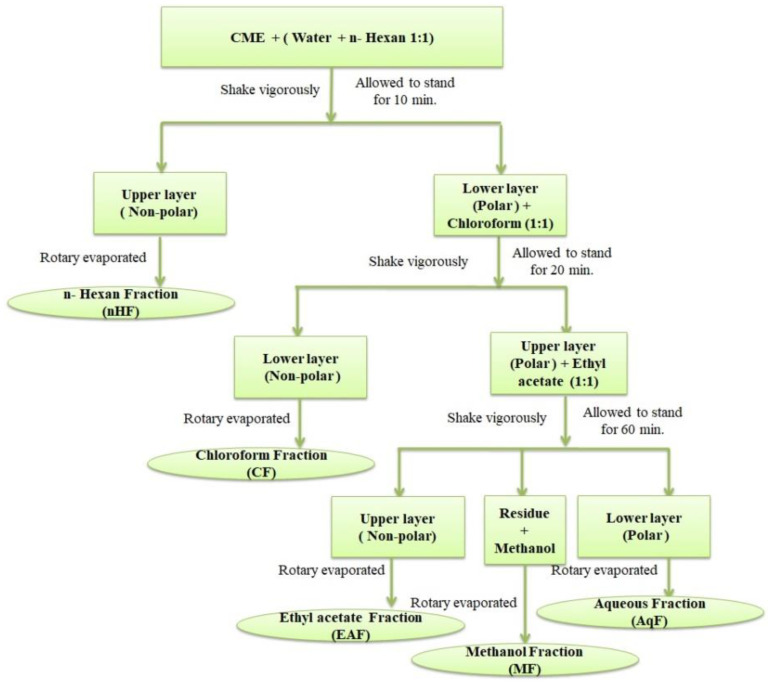
Fractionation scheme: a flow diagram representing the polarity-based solvent–solvent fractionation of crude methanolic extract (CME) of *Dryopteris ramosa*.

**Figure 3 antibiotics-10-01128-f003:**
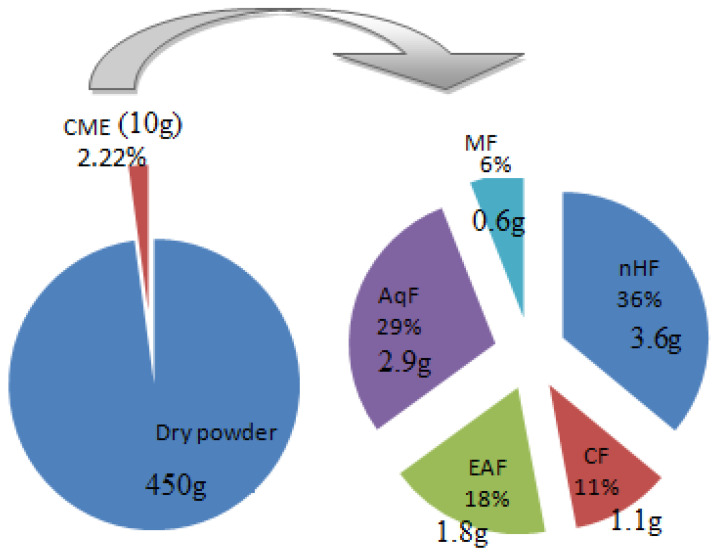
The percentage yield of various fractions obtained from 450 g dry weight of *Dryopteris ramosa*.

**Figure 4 antibiotics-10-01128-f004:**
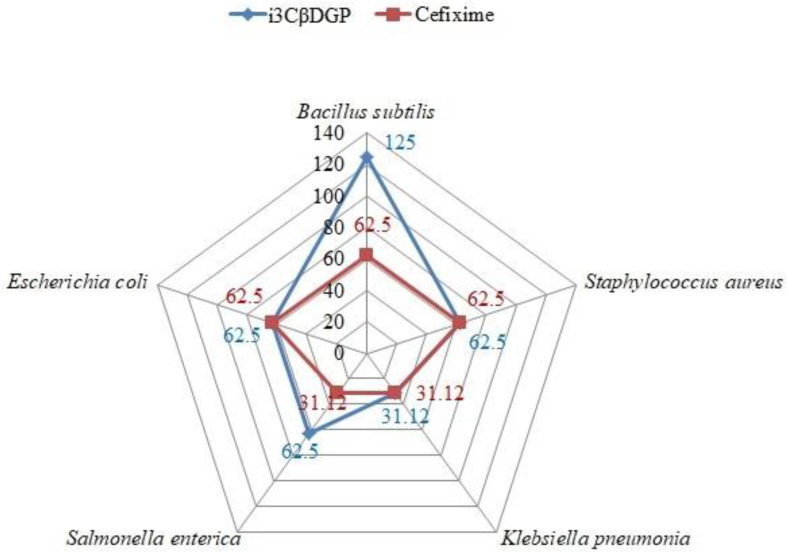
Comparison between MIC values (µg/mL) of iriflophenone-3-C-β-D-glucopyranoside (i3CβDGP) and Cefixime (well-known antibiotic) against various bacterial pathogens of human GIT.

**Figure 5 antibiotics-10-01128-f005:**
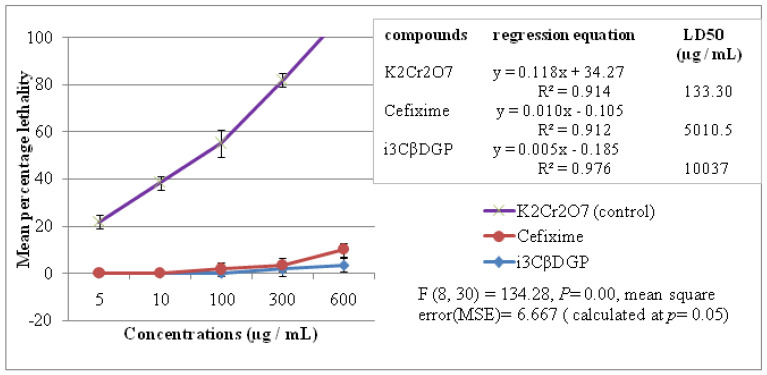
The cytotoxicity determination of iriflophenone-3-C-β D glucopyranoside (i3cβDGP) and its comparison with antibiotic (Cefixime) and control (K_2_Cr_2_O_7_). Data on top right corner show the LD_50_.

**Figure 6 antibiotics-10-01128-f006:**
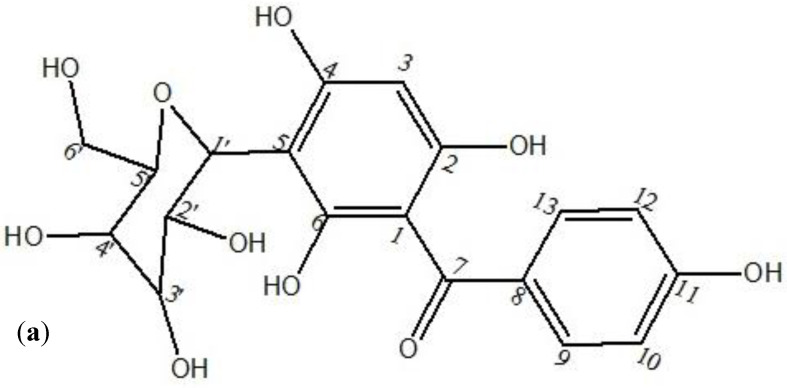
(**a**) Structural formula of “iriflophenone-3-C-β-D glucopyranoside” isolated from AqF of *D. ramosa* and (**b**) Proposed fragmentation scheme of iriflophenone-3-C-β-D glucopyranoside. Molecular ion is (M + H) 409 *m*/*z*. This scheme is based on fragments peaks observed for iriflophenone-3-C-β-D glucopyranoside during ESI-TOF/MS (positive ion mode).

**Figure 7 antibiotics-10-01128-f007:**
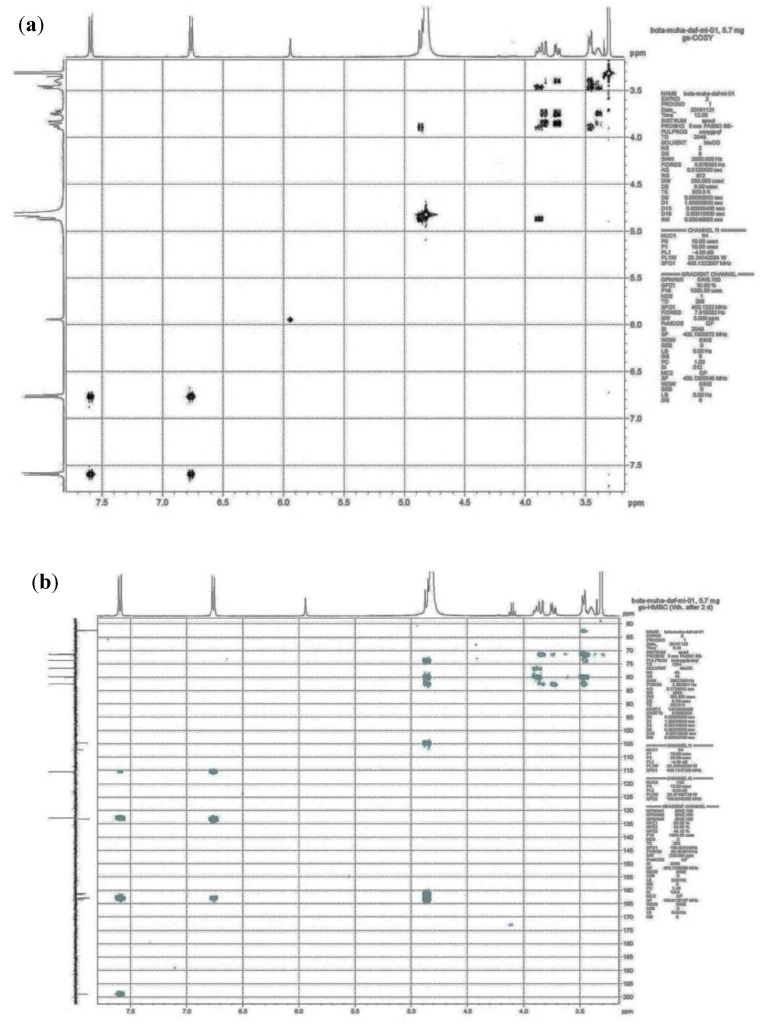
(**a**) gs-COSY spectra of “iriflophenone-3-C-β-D glucopyranoside” isolated from AqF of *D. ramosa* and (**b**) gs-HMBC spectra of “iriflophenone-3-C-β-D glucopyranoside” isolated from AqF of *D. ramosa*.

**Table 1 antibiotics-10-01128-t001:** Role of bacteria in GIT disorders in human.

Microorganism	Importance in GIT (Human)
*Bacillus subtilis*	Useful bacteria, in many animal species, feed supplementation with *Bacillus subtilis* used as prebiotics and probiotics [10]. Induce development of the gut-associated lymphoid tissue (GALT) and the pre-immune antibody repertoire [11].
*Staphylococcus aureus*	Produce staphylococcal enterotoxins (SEs), the causative agents of staphylococcal food poisoning [12].
*Klebsiella pneumoniae*	In Asia, it is the leading cause of pyogenic liver abscess (PLA) [13,14] The colonization of virulent type *K. pneumoniae* in intestine causes PLA [15]
*Salmonella enterica*	Gastroenteritis (diarrhea, abdominal cramps and fever) Enteric fevers, including typhoid fever [16]
*Escherichia coli*	Among the *E. coli* strains that can cause intestinal disease in humans, there are at least 6 well-characterized classes or pathotypes, as follows: enteropathogenic *E. pneumoniae* (EPEC), enterohemorrhagic *E.coli* (EHEC), enterotoxinogenic *E.coli* (ETEC), enteroaggregative *E. coli* (EAEC), enteroinvasive *E. coli* (EIEC), and diffusely adherent *E. coli* (DAEC) [17]

**Table 2 antibiotics-10-01128-t002:** Comparing the antibacterial potential of crude methanolic extract of *Dryopteris ramosa*, iriflophenone-3-C-β-D glucopyranoside and well-known antibiotic (Cefixime) against five bacterial strains.

	Conc. (µg/mL)	*Bacillus subtilis* ATCC 6633	*Staphylococcus aureus* ATCC 6538	*Klebsiella pneumoniae* ATCC 700603	*Salmonella enterica*. subsp. *enterica* Serovar Setubal ATCC 19196	*Escherichia coli* ATCC 25922
		Inhibition zone (in millimeters)				
**CME**	10	11 ± 0.7	12 ± 0.7	12 ± 0.3	11 ± 0.7	10 ± 0.7
	100	19 ± 0.3	24 ± 0.3	31 ± 0.3	22 ± 0.7	24 ± 0.7
	500	31 ± 0.7	38 ± 0.3	45 ± 0.7	43 ± 0.7	38 ± 0.7
	1000	43 ± 0.3	66 ± 0.7	65 ± 0.3	61 ± 0.3	54 ± 0.3
	**MIC** (µg/mL)	**250 ± 3.9**	**125 ± 7.2**	**62.5 ± 7.2**	**125 ± 3.9**	**125 ± 7.2**
**i3βDGP**	10	18 ± 0.3	23 ± 0.3	24 ± 0.7	19 ± 0.7	19 ± 0.7
	100	26 ± 0.7	32 ± 0.3	49 ± 0.3	32 ± 0.7	29 ± 0.7
	500	35 ± 0.3	45 ± 0.3	68 ± 0.3	55 ± 0.3	42 ± 0.7
	1000	48 ± 0.7	71 ± 0.7	85 ± 0.3	76 ± 0.7	55 ± 0.3
	**MIC** (µg/mL)	**125 ± 7.2**	**62.5 ± 7.2**	**31.1 ± 7.2**	**62.5 ± 14.4**	**62.5 ± 7.2**
**Cefixime ***	10	19 ± 0.3	23 ± 0.3	25 ± 0.7	26 ± 0.3	21 ± 0.3
	100	27 ± 0.7	31 ± 0.3	49 ± 0.3	36 ± 0.3	32 ± 0.3
	500	39 ± 0.7	44 ± 0.7	70 ± 0.3	59 ± 0.3	43 ± 0.7
	1000	51 ± 0.7	67 ± 0.3	94 ± 0.3	78 ± 0.3	56 ± 0.3
	**MIC** (µg/mL)	**62.5 ± 7.2**	**62.5 ± 7.2**	**31.1 ± 7.2**	**31.1 ± 7.2**	**62.5 ± 7.2**

n = 3, CME = crude methanolic extract of *Dryopteris ramosa*, i3CβDG = iriflophenone-3-C-β-D glucopyranoside, * control/standard, well-known antibiotic.

## Data Availability

The data presented in this study are available on request from the corresponding author.

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
