# Peer review of "Iriflophenone-3-C-β-d Glucopyranoside from Dryopteris ramosa (Hope) C. Chr. with Promising Future as Natural Antibiotic for Gastrointestinal Tract Infections"

_antibiotics, 2021, doi:10.3390/antibiotics10091128_

Round 1

Reviewer 1 Report

D. ramosa is used in a specific region from Pakistan in the treatment of gastrointestinal infections. The authors isolated its pure bioactive compound and tested its in vitro antibacterial and cytotoxic effect.

I have some questions. I have not found other studies evaluating the antibacterial activity from D. ramosa. So, I am speculating that its use is not popularly difused, maybe it is too restrict. What do you think about it?

The isolated compound which you have tested: there are studies about its biological activities? Or is it a totally novel compound?

Why have you used cefixime as the control?

The explanation about figures 5 and 6 need to be developed, they are so superficial and do not bring new information to the reader. For example, compare it to cefixime in the text, not only in the figure.

Author Response

Dated: 7th September 2021

Dear Editor,

Greetings,

Thank you very much for your time and comments regarding our manuscript (Antibiotics-1369151). Our manuscript “Iriflophenone -3-C-β-D Glucopyranoside from Dryopteris ramosa (Hope) C. Chr. With Promising Future as Natural Antibiotic for GIT Infections” has been revised carefully and here we are giving our response to the reviewers’ comments. We have improved the manuscript according to the reviewers’ comments and suggestions. All the revisions can be easily identified from manuscript highlighted with Blue color.

Once again thanks for your co-operation and valuable comments and suggestion. Moreover, the efforts of the reviewer are highly appreciated. We are hoping for pleasant response and further good comments (if any) from your side.

Dr. Abdul Qayyum

Department of Agronomy

The University of Haripur 22620 Pakistan

******************************************************************************

We are thankful to editor and reviewers for timely completion of review process and providing us with valuable feedback.

Response to Reviewer # 1:

Dear reviewer, we are grateful to you for your comments and suggestions for the improvement of our research manuscript. We have tried our best to revise the manuscript in light of your comments.

Comment:  D. ramosa is used in a specific region from Pakistan in the treatment of gastrointestinal infections. The authors isolated its pure bioactive compound and tested its in vitro antibacterial and cytotoxic effect.

The paper is easy to read and many of the discussions are sound and in agreement with existing works. Nonetheless, reviewer asked three questions and an explanation of figures presented in manuscript.

Comment:      Question 1: I have some questions. I have not found other studies evaluating the antibacterial activity from D. ramosa. So, I am speculating that its use is not popularly difused, maybe it is too restrict. What do you think about it?

Response:       Dryopteris ramosa is commonly found in northern areas of Pakistan especially in the Galiyat region of Pakistan. Many researchers have highlighted the medicinal importance of this plant species. In the present manuscript, the “3rd paragraph” in the “introduction section” (Line 14-22 introduction) clearly indicating these previous finding about D. ramosa including antibacterial potential of D. ramosa. Further, recently an article published in “BMC Complementary and alternative medicines” by Alam et al. (2021) 21:197, discussed the “Phytochemical, antimicrobial, antioxidant and enzyme inhibitory potential of medicinal plant Dryopteris ramosa (Hope) C. Chr.”

Comment:      Question 2:  The isolated compound which you have tested: there are studies about its biological activities? Or is it a totally novel compound?

Response:       The isolated compound “Iriflophenone-3-C-β-D glucopyranoside (i3βDGP)” was also isolated from Cyclopia subternata (honeybush) and some other plant species but this is the first time reported and isolated from fern genus “Dryopteris” and hence D. ramosa. In previous studies the “i3βDGP” was reported for its anti-hyperglycemic activities and antioxidant properties as already indicated in discussion section of this manuscript. (Please see Discussion, line 56-57). (Highlighted in blue colour)

Comment:      Question 3:  Why have you used cefixime as the control?

Response:       Cefixime antibiotic was used as control in the present study due to following reasons;

  1. Cefixime is a 4th generation antibiotics and has a broad range of action spectrum against Gram positive as well as against Gram negative bacteria.
  2. In the literature, cefixime was used as control in the considerable number of research articles which have been reported the antibacterial properties of either plants extracts or pure compound isolated from plants.
  • Cefixime is used as standard antibiotic in our lab because it is not very costly and is easily available in the market.

Comment:      Suggestion: The explanation about figures 5 and 6 need to be developed, they are so superficial and do not bring new information to the reader. For example, compare it to cefixime in the text, not only in the figure.

Response:       The explanation of the figure 5a and 5b have been added as suggested by the reviewer. Please see the Lines 17-34 in the discussion section of the article (Highlighted in blue colour).

Figure 6 have been compared in the test of manuscript as suggested by the reviewer. Please see the lines 50-53 in the discussion section. (Highlighted in blue colour).

******

Reviewer 2 Report

The authors are dealing with the isolation and identification the of bioactive compound from aqueous extract of the Dryopteris ramosa and its evaluation of antibacterial and cytotoxic potential. The elucidation of the isolated compound by MS and NMR were correctly performed. The isolation of the compound iriflophenone-3-C-β-D-glucopyranoside (2.3. and Figure 2) need more explanation. The authors claim to fractionate the crude methanolic extract (10 g). The last fraction is aqueous fraction. However, how is it possible that after adding methanol the whole extract was not dissolved, i.e. it dissolved in water, and thus the AqF extract (2 g) was obtained? Also, it would be good to add HPLC chromatogram and MS spectra (at least in supplement). Finally, English language should be read and corrected by native English speaker.

Other remarks:

Title:      Avoid abrev. in title – GIT. In addition, it seems bold to compare it to antibiotics based on in vitro studies. Also smaller letter D should be written i.e. in in the level of small letter “u”: Iriflophenone-3-C-β-D-Glucopyranoside

Line 14 Dryopteridaceae

Line 35   “1.  . INTRODUCTION” Dot should be erased i.e. “1. INTRODUCTION”. Please check all the title as this is erroneous in most cases

Line 43-47           correct English

Line 50  Dryopteridaceae

Line 59 GIT abrev. used for the first time – should be defined

Line 60 GIT complaints??

Line 62  in vitro

Line 66  human gastrointestinal tract (GIT) – should be defined earlier

Line 86  Please define the mass of the plant material used.

Line 92  “…was repeated three times.”

Line 131-135       plant D. ramosa and name of bacteria species should be italic

Line 239                “The cytotoxicity…”

Line 256                “…”of this plant is used…”;

Line 256-257       [5-7]

Line 259                correct English

Line 286                [27-29]

References are missing doi. Also, please arrange them according the guide for authors.

Author Response

Dated: 7th September 2021

Dear Editor,

Greetings,

Thank you very much for your time and comments regarding our manuscript (Antibiotics-1369151). Our manuscript “Iriflophenone -3-C-β-D Glucopyranoside from Dryopteris ramosa (Hope) C. Chr. With Promising Future as Natural Antibiotic for GIT Infections” has been revised carefully and here we are giving our response to the reviewers’ comments. We have improved the manuscript according to the reviewers’ comments and suggestions. All the revisions can be easily identified from manuscript highlighted with Red color.

Once again thanks for your co-operation and valuable comments and suggestion. Moreover, the efforts of the reviewer are highly appreciated. We are hoping for pleasant response and further good comments (if any) from your side.

Dr. Abdul Qayyum

Department of Agronomy

The University of Haripur 22620 Pakistan

******************************************************************************

We are thankful to editor and reviewers for timely completion of review process and providing us with valuable feedback.

Response to Reviewer # 2:

Dear reviewer, we are grateful to you for your comments and suggestions for the improvement of our research manuscript. We have tried our best to revise the manuscript in light of your comments.

Comment:      The authors are dealing with the isolation and identification the of bioactive compound from aqueous extract of the Dryopteris ramosa and its evaluation of antibacterial and cytotoxic potential. The elucidation of the isolated compound by MS and NMR were correctly performed. The isolation of the compound iriflophenone-3-C-β-D-glucopyranoside (2.3. and Figure 2) need more explanation.

Response:       Adequate explanation regarding figure 2 has been added in section 2.3 of the manuscript as suggested by the reviewer. (Highlighted in Red colour)

Comment:      The authors claim to fractionate the crude methanolic extract (10 g). The last fraction is aqueous fraction. However, how is it possible that after adding methanol the whole extract was not dissolved, i.e. it dissolved in water, and thus the AqF extract (2 g) was obtained?

Response:      The authors has fractionated the crude methanolic extract (10g) based on solvent-solvent polarity based in the order of decreasing polarity of the solvents i.e n-Hexan< Chloroform<Ethylacetae <Water. When crude methanolic extract was dissolved in n-hexan along with water, only those constituents of CME were dissolved in n-hexans which have polarity more or less equal to n-hexan. As the density of n-hexan is less than water, therefore, n-Hexan form the upper layer. Other fractions were obtained as described in section 2.3 of the manuscript and figure 2 based on the same concept.

Moreover, regarding the last step of fractionation, when ethyl acetate was mixed with polar layer, after an hour, the fractionation flask has three components; (i) upper soluble polar layer (ii) lower soluble non-polar layer and (iii) insoluble part. All the three components of the flask were separated. EAF was obtained from upper polar layer while the AqF was obtained from lower non-polar layer after rotary drying. The insoluble part was dissolved in methanol and after rotary evaporation; it was label as “MF”.

Comment:      Also, it would be good to add HPLC chromatogram and MS spectra (at least in supplement).

Response:       Provided as supplementary files

Comment:      Finally, English language should be read and corrected by native English speaker.

Response:       Corrected with best of our abilities as suggested by the reviewer.

Other Remarks:

Comment:      Avoid abrev. in title – GIT. In addition, it seems bold to compare it to antibiotics based on in vitro studies. Also smaller letter D should be written i.e. in in the level of small letter “u”: Iriflophenone-3-C-β-D-Glucopyranoside

Response:       Modified as suggested by the reviewer (Highlighted in Red colour)

Comment:      Line 14 Dryopteridaceae

Response:       Corrected as suggested by the reviewer (Highlighted in Red colour)

Comment:      Line 35   “1.  . INTRODUCTION” Dot should be erased i.e. “1. INTRODUCTION”. Please check all the title as this is erroneous in most cases

Response:       Corrected as suggested by the reviewer

Comment:       Line 43-47           correct English

Response:       These lines have been modified as suggested by the reviewer in the revised manuscript. Line 44-47 (Highlighted in Red colour)

Comment:      Line 50  Dryopteridaceae

Response:       Corrected as suggested by the reviewer (Highlighted in Red colour)

Comment:      Line 59 GIT abrev. used for the first time – should be defined

Response:       abbreviation is defined, line 58 (Highlighted in Red colour)

Comment:      Line 60 GIT complaints??

Response:       GIT complaints are replaced with GIT disorders. Line 60 (Highlighted in Red colour)

Comment:      Line 62  in vitro

Response:       Corrected, Line 62, (Highlighted in Red colour)

Comment:      Line 66 human gastrointestinal tract (GIT) – should be defined earlier

Response:       Defined and Highlighted in Red colour, line 66

Comment:      Line 86  Please define the mass of the plant material used.

Response:       The mass of collected plant was added in line 86 and Highlighted in Red colour

Comment:      Line 92  “…was repeated three times.”

Response:       Modified as suggested by the reviewer, Line 112, (Highlighted in Red colour)

Comment:      Line 131-135       plant D. ramosa and name of bacteria species should be italic

Response:       Corrected as suggested by the reviewer, lines 151-155. (Highlighted in Red colour)

Comment:      Line 239                “The cytotoxicity…”

Response:       Lines 279, Corrected, (Highlighted in Red colour)

Comment:      Line 256                “…”of this plant is used…”;

Response:       Name of the plant is added in revised manuscript. Line 288. (Highlighted in Red colour)

Comment:      Line 256-257       [5-7]

Response:       Corrected, line 290, (Highlighted in Red colour)

Comment:      Line 259                correct English

Response:       Line 292 in revised manuscript, the sentence has been rephrased as per your valuable suggestion (Highlighted in Red colour)

Comment:      Line 286                [27-29]

Response:       Line 331, corrected, (Highlighted in Red colour)

Comment:      References are missing doi. Also, please arrange them according the guide for authors

Response:       References has been arranged according to the author’s guideline.

Supplementary Figure 1. MS of isolated compound “Iriflophenone-3-C-β-D glucopyranoside”  from AqF of Dryopteris ramosa. Molecular ion at M+ 409m/z.

Supplementary Figure 2. HPLC chromatogram showing peak at 6.467 minutes and UV-Vis absorption spectra of isolated compound (Iriflophenone-3-C-β-D glucopyranoside) from AqF of Dryopteris ramosa.

Reviewer 3 Report

The manuscript deals with a very important topic because of the urgent need of new antimicrobial agents. However, the scientific soundness of the research study requires some improvement.

I think that the antimicromicrobial activity of the AqF should be tested and compared with the activity of the isolated compound. It is generally said that plant extracts have a remarkable potential compared to single molecules because of the synergistic effect between bioactive compounds. In this frame, it is crucial to investigate the role of i3CβDGP within the fraction.

Is the i3CβDGP the only compound isolated from the AqF of D. ramosa?

In addition, experiments aimed to elucidate the mechanism of action of i3CβDGP has to be performed.

"Authentic strains" is referred to "reference strains"?

It is hard to believe that 20 microL of volume could spread when spotted into a 8 mm well. It seems a small amount of sample not reaching the edge of the agar-well.

Author Response

Dated: 7th September 2021

Dear Editor,

Greetings,

Thank you very much for your time and comments regarding our manuscript (Antibiotics-1369151). Our manuscript “Iriflophenone -3-C-β-D Glucopyranoside from Dryopteris ramosa (Hope) C. Chr. With Promising Future as Natural Antibiotic for GIT Infections” has been revised carefully and here we are giving our response to the reviewers’ comments. We have improved the manuscript according to the reviewers’ comments and suggestions. All the revisions can be easily identified from manuscript highlighted with Green color.

Once again thanks for your co-operation and valuable comments and suggestion. Moreover, the efforts of the reviewer are highly appreciated. We are hoping for pleasant response and further good comments (if any) from your side.

Dr. Abdul Qayyum

Department of Agronomy

The University of Haripur 22620 Pakistan

***************************************************************************

We are thankful to editor and reviewers for timely completion of review process and providing us with valuable feedback.

Response to Reviewer # 3:

Dear reviewer, we are grateful to you for your comments and suggestions for the improvement of our research manuscript. We have tried our best to revise the manuscript in light of your comments.

Comment:      The manuscript deals with a very important topic because of the urgent need of new antimicrobial agents. However, the scientific soundness of the research study requires some improvement.

I think that the antimicromicrobial activity of the AqF should be tested and compared with the activity of the isolated compound. It is generally said that plant extracts have a remarkable potential compared to single molecules because of the synergistic effect between bioactive compounds. In this frame, it is crucial to investigate the role of i3CβDGP within the fraction.

Response:       Line 334-345 in revised manuscript. The antibacterial potential of crude extract and water fraction of Dryopteris ramosa was discussed and Highlighted in Green colour.

Comment:      Is the i3CβDGP the only compound isolated from the AqF of D. ramosa?

Response:       Yes, for the present article it’s the only compound isolated.

Comment:      In addition, experiments aimed to elucidate the mechanism of action of i3CβDGP has to be performed.

Response:       This research artical is based on my Ph. D work. Unfortunately, the mechanism of action was not in my Ph. D proposal. Therefore, it was not considered. But in future we will focus on this aspect as well.

Comment:      Authentic strains" is referred to "reference strains"?

Response:       This mean the bacterial strain were properly identified by an expert microbiologist and there is no ambiguity about these strains that were used in the present study.

Comment:      It is hard to believe that 20 microL of volume could spread when spotted into a 8 mm well. It seems a small amount of sample not reaching the edge of the agar-well.

Response:       Agar well diffusion method is widely used to determine the antibacterial potential of botanicals. Majority of the researchers dealing with Agar well diffusion method used this (20µL) volume of tested sample in 8mm well. In our practical experience, this volume although sees very little but able to diffuse through the agar and showed inhibition zone (depends upon its antibacterial potential). 

********

Round 2

Reviewer 3 Report

Once again, the Authors did not pay enough attention to bacterial nomenclature. Klebsiella pneumonia and Escherichia colli are wrong. Samonella Setubal could be changed into Salmonella enterica subsp. enterica serovar Setubal. Are the Authors referring to this strain?

In response to my comment "Authentic strains" is referred to reference strains? the Authors confused me even more. Are bacterial strains purchased by ATCC, as they stated in paragraph 2.6.1, or strains have been isolated by an expert microbiologist, as they wrote in their response? ATCC strains do not require neither isolation nor identification procedures by experts. There can be no ambiguity by using ATCC strains. Otherwise, identification is required when clinical samples are used.

The antimicrobial activity of the crude extract should be evaluated in the present study. The Authors report data from literature obtained testing bacterial strains different from those used herein. The comparison should be carried out by using similar experimental conditions.

In my opinion, some empirical evidence of the mechanism of action should be provided.

Author Response

Dated: 12th September 2021

Dear Editor,

Greetings,

Thank you very much for your time and comments regarding our manuscript (Antibiotics-1369151). Our manuscript “Iriflophenone -3-C-β-D Glucopyranoside from Dryopteris ramosa (Hope) C. Chr. With Promising Future as Natural Antibiotic for GIT Infections” has been revised carefully and here we are giving our response to the reviewers’ comments. We have improved the manuscript according to the reviewers’ comments and suggestions. All the revisions can be easily identified from manuscript highlighted with Green color.

Once again thanks for your co-operation and valuable comments and suggestion. Moreover, the efforts of the reviewer are highly appreciated. We are hoping for pleasant response and further good comments (if any) from your side.

Dr. Abdul Qayyum

Department of Agronomy,

The University of Haripur 22620 Pakistan

***************************************************************************

We are thankful to editor and reviewers for timely completion of review process and providing us with valuable feedback.

Response to Reviewer # 3:

Dear reviewer, we are grateful to you for your comments and suggestions for the improvement of our research manuscript. We have tried our best to revise the manuscript in light of your comments.

Comment:      Once again, the Authors did not pay enough attention to bacterial nomenclature. Klebsiella pneumonia and Escherichia colli are wrong.

Response:       The bacterial nomenclature, Klebsiella pneumoniae and Escherichia coli have been corrected throughout the manuscript. All the corrections have been highlighted in green colour.

Comment:      Salmonella Setubal could be changed into Salmonella enterica subsp. enterica serovar Setubal. Are the Authors referring to this strain?

Response:       Yes, Samonella setubal is the heterotypic synonym of Salmonella enterica subsp. enterica serovar Setubal. The name has been changed in the manuscript as suggested by the reviewer. Highlighted in Green colour.

Comment:      In response to my comment "Authentic strains" is referred to reference strains? the Authors confused me even more. Are bacterial strains purchased by ATCC, as they stated in paragraph 2.6.1, or strains have been isolated by an expert microbiologist, as they wrote in their response? ATCC strains do not require neither isolation nor identification procedures by experts. There can be no ambiguity by using ATCC strains. Otherwise, identification is required when clinical samples are used.

Response:       We felt sorry that our previous response confused the reviewer. Respected sir, “Authentic strain” means “ATCC bacterial strain” which were used in the present study.

Comment:      The antimicrobial activity of the crude extract should be evaluated in the present study. The Authors report data from literature obtained testing bacterial strains different from those used herein. The comparison should be carried out by using similar experimental conditions.

Response:       The antibacterial potential of Crude methanolic extract of D. ramosa was presented in table 2 in the revised manuscript. These results were compared in discussion part of the revised manuscript (discussion-Lines 57-64). Highlighted in Green colour.

Comment:      In my opinion, some empirical evidence of the mechanism of action should be provided.

Response:       The empirical mechanism of action has been incorporated in the discussion section of the manuscript. Page 12-13, discussions lines 53-56. Highlighted in Green colour.

********
